# Molecular Genetics Overview of Primary Mitochondrial Myopathies

**DOI:** 10.3390/jcm11030632

**Published:** 2022-01-26

**Authors:** Ignazio Giuseppe Arena, Alessia Pugliese, Sara Volta, Antonio Toscano, Olimpia Musumeci

**Affiliations:** 1Unit of Neurology and Neuromuscular Disorders, Department of Clinical and Experimental Medicine, University of Messina, 98125 Messina, Italy; igna.arena@live.it (I.G.A.); alessiapugliese76@gmail.com (A.P.); atoscano@unime.it (A.T.); 2Department of Neurosciences, University of Padova, 35100 Padova, Italy; sara.volta@studenti.unipd.it

**Keywords:** mitochondrial myopathy, exercise intolerance, ophtalmoplegia, mtDNA, nDNA, oxidative phosphorylation

## Abstract

Mitochondrial disorders are the most common inherited conditions, characterized by defects in oxidative phosphorylation and caused by mutations in nuclear or mitochondrial genes. Due to its high energy request, skeletal muscle is typically involved. According to the International Workshop of Experts in Mitochondrial Diseases held in Rome in 2016, the term Primary Mitochondrial Myopathy (PMM) should refer to those mitochondrial disorders affecting principally, but not exclusively, the skeletal muscle. The clinical presentation may include general isolated myopathy with muscle weakness, exercise intolerance, chronic ophthalmoplegia/ophthalmoparesis (cPEO) and eyelids ptosis, or multisystem conditions where there is a coexistence with extramuscular signs and symptoms. In recent years, new therapeutic targets have been identified leading to the launch of some promising clinical trials that have mainly focused on treating muscle symptoms and that require populations with defined genotype. Advantages in next-generation sequencing techniques have substantially improved diagnosis. So far, an increasing number of mutations have been identified as responsible for mitochondrial disorders. In this review, we focused on the principal molecular genetic alterations in PMM. Accordingly, we carried out a comprehensive review of the literature and briefly discussed the possible approaches which could guide the clinician to a genetic diagnosis.

## 1. Introduction

According to the International Workshop of Experts in Mitochondrial Diseases held in Rome in 2016, Primary Mitochondrial Myopathies (PMMs) can be defined as disorders that lead to defects in oxidative phosphorylation (OXPHOS) and that mainly, but not exclusively, affect skeletal muscle [1]. Progressive external ophtalmoplegia (PEO), eyelid ptosis, exercise intolerance and muscle weakness are the most common symptoms of myopathy that occur in mitochondrial diseases. Myopathy can be isolated but more frequently is associated with other clinical manifestations [2].

PMM are mostly the expression of genetic molecular alterations that may involve mitochondrial DNA (mtDNA) or nuclear DNA (nDNA), or may be due to an impairment of the intergenomic communications [3].

In recent years, there was a spread of new technologies, such as Next-Generation Sequencing (NGS), leading to the discovery and individualization of more and more mutations. To date, more than 350 genes in both mitochondrial and nuclear genomes are known to cause primary mitochondrial diseases [4].

In the present review we will give an overview of the principal molecular genetic defects linked to PMM (Table 1). We focus searched PubMed for articles published in English from 1 January 2011 to 1 November 2021, using the search terms “mitochondrial disease OR disorder”, “CPEO”, “myopathy”, “exercise intolerance”, “genetic”, “mtDNA”, “nDNA”. We further evaluated the reference lists from relevant articles including recent reviews (Table 2) and case series reports

## 2. General Aspects

Each mitochondrion has its own genetic code, constituting circular, double-stranded DNA of 16,569 base pairs. The mtDNA as whole contains 37 genes. Of these, there are 24 genes which code for molecules essential for the synthesis of the protein subunits of the respiratory chain complexes. Particularly, these genes encode for two ribosomal RNA, 12s and 16s rRNA, and 22 transfer RNA (tRNA); The remaining 13 genes encode for 13 messenger RNA (mRNA) which are translated into 13 polypeptide subunits involved in the mitochondrial respiratory chain (MRC). Specifically, seven of these genes (MT-ND1, MT-ND2, MT-ND3, MT-ND4, MT-ND4L, MT-ND5, MT-ND6) encode for subunits of complex I; one gene encodes for cytochrome b (MTCYB), which represents a subunit of complex III; three genes encode for cytochrome I, II, III, which are part of the complex IV (MT-CO1, MT-CO2, MT-CO3), and finally two genes encode for ATPases 6 and 8, subunits of complex V (MT-ATP6, MT-ATP8) [17]. However, only 15% of mitochondrial proteins are encoded by the mitochondrial genome (only 13 subunits among the 90 proteins of the MRC subunits) since the majority is nDNA encoded [18].

Usually, all the mitochondrial genetic material is inherited by maternal lineage since the mitochondria contained in the spermatozoa do not enter the egg cell and are eliminated through different mechanism [19]. Consequently, even if the mtDNA is transmitted by the mother to the sons (males and females) without distinction, only the daughters pass it on to the next generation. Paternal mtDNA does not contribute to mitochondrial inheritance, although there are rare cases reporting some sperm mitochondria penetrating the egg [20]. Additionally, an mtDNA-linked disease could be an expression of a sporadic alteration, which is what usually happens in single, large-scale mtDNA deletion syndromes [5,6].

Another important feature of mitochondrial genetics is the so-called threshold effect, which determines the phenotypic expression. In fact, there are up to thousands of mtDNA copies in each cell. Despite the rare cases of mutant homoplasmia (condition in which the mitochondrial genotype of a subject is composed of a single normal or altered type of mtDNA), wild-type unmutated genome and mutant DNA usually coexist within the same cell (heteroplasmy) [21]. In this latter situation, the mitochondrial genetic material is randomly distributed to the daughter cells at the time of mitosis. Accordingly, there will be a dysfunction only when the copies of the mutated gene accumulate above a certain threshold, and that is when the damaging effect of the mutation will no longer be compensated by the coexisting normal mtDNA. Moreover, the phenotypic expression of the disease, in terms of severity, is strongly conditioned by the amount of mutated genome present in the cells of a tissue, which could vary even through the years [22].

Interestingly, there is no genotype–phenotype correlation between the site of the mutation and the clinical phenotype. Conversely, some genes are most frequently associated with some specific disease known as “classic mitochondrial syndromes” such as the MT-TL1 gene mutations associated with mitochondrial encephalopathy, lactic acidosis, and stroke-like episodes (MELAS), or the MT-TK mutation correlated with myoclonic epilepsy with ragged red fibers (MERRF), ND genes alterations with Leber hereditary optic neuropathy (LHON), and MT-CYB gene with exercise intolerance [23,24,25,26]. On the other side, nDNA defects recognize a Mendelian inheritance, thereby are commonly transmitted to the offspring in an autosomal dominant (ad) or recessive (ar), or more rarely, X-linked fashion [27].

Mitochondria are involved in various cellular functions, the most important of which is certainly oxidative phosphorylation (OXPHOS), which represents the final stage of cellular respiration, after glycolysis, oxidative decarboxylation of pyruvate and the Krebs cycle [28]. Thereby, most of the mutations affecting mitochondria, either nDNA or mtDNA, somehow compromise this crucial and final stage of the cellular metabolism. Given this background and because its high energy-demand, skeletal muscle tissue is one of the most affected tissues.

## 3. Defects of MRC Complexes

### 3.1. Complex I and Assembly Factors

Complex I (CI), also known as NADH-coenzyme Q oxidoreductase, is the largest multimeric enzyme complex of the electron transport chain (ETC) and it is composed of 45 structural subunits. MtDNA encodes for seven subunits, while the remaining 38 proteins and ancillary factors have a nuclear origin [7]. To date, 15 factors with a role in the assembling of complex I have been found [29].

Leigh syndrome or Leigh-like syndrome are the most frequent clinical expressions of CI deficiency [30]. Nevertheless, the phenotypes documented are extremely wide, including LHON, MELAS, and exceptionally, cases of dystonia and ataxia. Muscle involvement is occasionally isolated, and it is usually part of a multisystem phenotype. Behind this last situation, there is generally a nuclear defect, while a clinical isolated myopathy is usually an expression of mitochondrial gene alterations [8].

Basically, almost every mtDNA gene coding for CI subunit is documented to be potentially involved in PMM (Table 3).

Mutations affecting MT-ND1 have been described both in homoplasmic and heteroplasmic form: for example, Rafiq J et al. described a family carrying a new homoplasmic mtDNA m.4087A>G mutation in the ND1 gene (MT-ND1), associated with isolated myopathy, recurrent episodes of myoglobinuria, and rhabdomyolysis [31]; conversely, Gorman GS et al. reported two novel heteroplasmic mutations (m.3365T>C predicting p.Leu20Pro and m.4175G>A predicting p.Trp290*) in the MT-ND1 gene of two different adults with considerable fatigue and dyspnea induced by progressive exertion, persistent hyperlactatemia and severe muscle-restricted, isolated CI deficiency [32].

The clinical pictures reported are usually remarkably similar, with patients often presenting with exercise intolerance, muscle weakness and sometimes with high level of lactate at rest: an MT-ND2 heteroplasmic mutation (m.4831G>A), consisting of a transition of the p.Gly121Asp in the ND2 protein, was described by Zanolini A et al. in a young patient complaining about exertion-related muscle weakness and lactic acidosis [33], and similar cases are reported as consequences of MT-ND4 and MT-ND5 point mutations [34,35].

Moreover, it is worth noting that most of the MT-ND1-ND6 genetic alterations are mostly the expression of transition occurring in the nucleotide sequences, conducing often to missense mutations and more rarely to nonsense mutations, although other different types of mutational mechanisms have been reported. In 2000, Musumeci O et al. individuated a pathogenic inversion of seven nucleotides altering three amino acids in a highly conserved region of the MT-ND1 gene in a patient presenting with exercise intolerance and myalgia since childhood [36]. A single nucleotide deletion in the MT-ND5 gene (m.12425delA), as a de novo mutational event, was reported by Alston et al. in a girl struggling with renal failure, myopathy, and long-lasting lactic acidosis. In this case, the truncation, which was heteroplasmic, led to a frameshift in the codon 30, with a change in the ND5 protein [37]. Finally, Yi Shiau et al. have more recently reported the cases of two patients carrying, respectively, a different frameshift alteration in MT-ND6 (m.14512_14513del) and a maternally inherited transversion in MT-ND1; particularly, the first proband was affected by a progressive exercise intolerance and mild muscular weakness since adolescence, while the second one suffered from a reduction in muscle strength in the hip flexion (MRC grade 4+/5) only after having experienced loss of vision, headache, deafness, vertigo, and sensory disturbances [38].

Furthermore, nDNA defects are more frequently responsible for complex I deficiency than mtDNA [52]. Most complex I diseases secondary to nDNA genes alterations are inherited in an autosomal recessive manner, although two are X-linked (NDUFA1 and NDUFB11) [53,54]. Nuclear pathological variants tend to be sporadic, and private mutations are described in few individuals or single families [55].

Nevertheless, according to some authors, more than half of complex I deficiencies are believed to be caused by mutations in ancillary factors necessary for proper complex I assembly and functioning, although relatively few patients have been described to have mitochondrial disease secondary to a mutation in a complex I assembly factors [56]. According to the literature, the principal assembly factors complex I encoding-genes leading to skeletal muscle involvement are TMEM126B, ACAD9 and NDUFAF3.

Cases of patients referring with exertion-caused progressive myalgia and presenting with muscle weakness in lower limbs at neurological examination with hyperlactatemia at rest were found to be associated with biallelic mutations in TMEM126B. Interestingly, most of these patients were carrying at least a mutation in the c.635G>T (pGly212Val), as described by two different works [39,40].

Likewise, exercise intolerance, muscle weakness and acidosis can be part of the complex I deficiency caused by ACAD9 mutations. ACAD9 is a gene expressed on chromosome 3 that codes for acyl-CoA dehydrogenase 9, located in the mitochondrion and required for the assembly of the mitochondrial complex I. Most of these changes are transmitted in an autosomal recessive way, and lead (especially in children) to multisystem pictures with cardiac involvement (hypertrophic cardiomyopathy) and other neurological symptoms (ataxia, dystonia) [57].

There are up to 70 cases of which most are missense mutations, although frame shift, nonsense, splice sites and initiation codon alterations are described. Particularly, in a cohort study by Repp et al., most of the patients presented in the first year of life, with 50% not managing to survive the first 2 years. Patients with a later presentation had longer life expectancy (more than 90% surviving the first 2 years). Cardiomyopathy (85%), muscular weakness (75%), exercise intolerance (72%), were the most frequent clinical features [41].

Similarly, NDUFAF3 was found to be mutated in association with a spectrum of severe phenotypes with complex I deficiency. Three cases are reported in the literature, being characterized by rapidly progressive syndrome with muscle hypertonia or macrocephaly with severe muscle weakness or myoclonic epilepsy and leukomalacia [58].

### 3.2. Complex II and Assembly Factors

Complex II (also called succinate-coenzyme Q oxidoreductase) is the only entirely nuclear-coded complex of the OXPHOS system. It is flavoprotein constituting four subunits (SDHA, SDHB, SDHC, SDHD), and has the double function of metabolizing succinate to fumarate (in the Krebs cycle) as well to transfer electrons from FADH2 to reduce ubiquinone to ubiquinol [59].

Although complex II deficiencies are exceedingly uncommon (almost 2–8% of mitochondrial disease cases), either homozygous or heterozygous pathogenic variants are reported in SDHA, SDHB, and SDHD genes, leading to some primary mitochondrial disease such as Leigh or Leigh-like syndrome [60]. Other organ systems such as the heart, muscle, and eyes are involved in about 50% of the cases, and interestingly, cardiomyopathy is associated with high mortality and morbidity [59].

Generally, these disorders are less severe when occurring in adults, while children often struggle with massive cognitive impairment, multiorgan involvement or even death in some cases.

SDHA, SDHB and SDHD mutations are inherited through a recessive manner and may lead to leukodystrophy and to isolated mitochondrial complex II deficiency [61].

Several complex II assembly factors have also been identified, including SDHAF2 and SDHAF1, which have been associated with autosomal recessive complex II deficiency and leukoencephalopathy [62].

### 3.3. Complex III and Assembly Factors

Complex III, ubiquinone–cytochrome c oxidoreductase, is composed of 11 subunits of which only one (cytochrome b) is encoded by the mitochondrial genome. Cytochrome B (MT-CYB), along with cytochrome c1 (CYC1) and the Rieske protein (UQCRFS1) represent the catalytic center. CYC1 as well the other structural subunits (UQCRB, UQCRQ, UQCRC2) and the assembly factors (BCS1L, LYRM7, UQCC2, and UQCC3) are coded by nDNA [63].

Functionally, complex III represents the center of different metabolic pathways as well as the MRC. In recent years, several new pathogenic variants have been associated with complex III dysfunction, involving both structural subunits and assembly factors with extremely heterogeneous clinical phenotypes (including hypoglycemia and hyperglycemia, hepatomegaly, renal tubular acidosis, Leigh Syndrome, and other neurological abnormalities) [64].

Myopathic involvement is reported in the form of exercise intolerance (sometimes associated with proximal limb weakness and myoglobinuria) in more than 50% of patients with mutation of the mitochondrial MT-CYB gene, coding for cytochrome b [9].

The first reported cases of MT-CYB mutations can be traced back to the 1990s, although the correlation between CIII deficiency and muscle involvement was known for some time before.

In fact, as early as the 1970s, Spriro AJ et al., reported the cases of a 46-year-old man and his son presenting muscular weakness and progressive ataxia [65]. Afterwards, several other cases were described: Darley-Usmar et al. presented a patient with lactic acidosis and muscle weakness while Hayes et al. reported the case of a Chilean girl with ptosis and fatigue [66,67]. A progressive exercise intolerance with a low CIII activity was also described [68].

However, only starting from the 1990s with mitochondrial DNA sequencing, was it possible to highlight the molecular basis of these disorders.

In 1996, Dumoulin R et al. identified the first cyt b missense pathogenic mutation in a young man with exercise intolerance. The alteration, which was heteroplasmic, consisted of the transition of guanine to adenine in position 15,615 of mtDNA, leading to the substitution Gly290Asp [42].

Particular importance must be given certainly to Andreu’s et al. study, with the description of five cases of patients with severe exercise intolerance, lactic acidosis present at rest (in four out of five patients), and biochemical evidence of complex III dysfunction. Of these patients, there were three nonsense mutations (G15084A, G15168A, and G15723A), one missense mutation (G14846A), and a 24-base deletion (from nucleotides 15,498 to 15,521) in cytochrome b. All the point mutations reported involved the substitution of adenine for guanine, but all on different locations. Moreover, there was no maternal inheritance, and there were no other mutations in other tissue districts beside muscle, suggesting that the disorder was due to somatic mutations in myogenic stem cells after germline differentiation [22].

Since then, new clinical phenotypes linked to MT-CYB have been identified such ataxia and MELAS syndrome, while a four-base deletion was individuated within a form of parkinsonism [69].

Furthermore, other forms of predominantly myopathic involvement have been reported. A stop-codon mutation (G15242A) of the mitochondrial encoded gene for cytochrome b predicting the loss of the last 215 amino acids of cytochrome b was identified [43].

Wibrand F et al. described a heteroplasmic mutation changing in a highly conserved region tyrosine to cysteine at position 278 in a patient with severe exercise intolerance in the context of a multisystem disorder including deafness, mental retardation, retinitis pigmentosa, cataract, growth retardation and epilepsy [70].

Schuelke M et al. described a 25-year-old patient with septo-optic dysplasia, retinitis pigmentosa, weakness, and hypertrophic cardiomyopathy since childhood, who carried the heteroplasmic mutation m.14849T>C predicting the p.Ser35Pro substitution within cyt b. [71].

Legros F et al. reported two different heteroplasmic mutations (substitutions p.Trp135Ter and p.Ser151Pro), in the muscle of two unrelated patients presenting with an effort-induced fatigability from their late childhood with a CIII activity below 20% in muscle, associated with a reduced amount of cyt b protein [72].

Bruno C et al. described a 40-year-old woman developing progressive exercise intolerance, lactic acidosis and muscle cramps beginning at the age of 30 years, because of mutation m.15170G>A leading to a stop codon (p.Gly142Ter) [44].

Another heteroplasmic mutation (m.15761G>A) leading to a stop codon (cyt b p.Gly339Ter) and thereby to a truncated protein was reported by Mancuso et al. in a 19-year-old woman suffering from exercise intolerance, vomiting and lactic acidosis [73].

Another nonsense mutation m.15800C>T (substitution p.Gln352Ter) leading to a truncation in cyt b protein is described by Lamantea E at al., in a 24-year-old woman who had exercise intolerance with muscle cramps and lactic acidosis [45].

Furthermore, nuclear CIII deficiencies are caused by recessively inherited mutations affecting nDNA genes encoding for structural subunits or assembly factors, and are associated with a wide range of clinical presentations and in some cases reduced CIII activity/amount in cultured fibroblasts or skeletal muscle [64].

Mutations in the BCS1L gene are reported as a cause of complex III dysfunction. BCS1L is a gene situated on chromosome 2q35, and is involved in the synthesis of an AAA protein (ATPase associated with diverse cellular activities) which plays a role in iron homeostasis and in the assembly of complex III, particularly in the incorporation of UQCRFS1. The clinical spectrum varies from purely visceral syndrome (such as GRACILE syndrome or Bjorstand syndrome) to a pure form of encephalopathy [10,64]. In these conditions, muscle involvement is mostly limited to only being a part of a multiorgan disorder.

### 3.4. Complex IV and Assembly Factors

Complex IV, or cytochrome c oxidase (COX), is situated in the inner mitochondrial membrane and consists of 14 structural subunits, since NDUFA4, previously assigned to complex I, was recently added as a new peripheral subunit of COX because NDUFA4 deficiency results in a severe loss of COX activity and represents a stoichiometric component of the individual COX complex [74]. COX consists of three catalytic subunits encoded by mt-DNA, MT-CO1, MT-CO2 and MT-CO3. The remaining subunits (COX4, COX5A, COX5B, COX6A, COX6B, COX6C, COX7A, COX7B, COX7C, COX8 and NDUFA4) and ancillary factors (COX10, COX14, COX 15, COX20, COA3, COA5, COA6, COA7, COA8, PET100, PET117, SCO1, SCO2 and SURF1) are instead entirely encoded by nDNA. COX functions as a dimer, with two copper binding sites, two heme groups, a magnesium ion, and a zinc ion. Remarkably, there are multiple tissue-specific isoforms for numerous subunits of COX (for example COX4, COX6A, COX6B, COX7A, COX8) [11].

As with other complex deficiencies, pathogenic mutations of COX have been associated with different clinical manifestation, ranging from isolated myopathy to severe multisystem disorder [75].

In 1999, Rahman S et al. identified the first missense mutation in the mtDNA gene for subunit II of cytochrome c oxidase (COX) in a 14-year-old boy with a proximal myopathy and lactic acidosis: particularly, mtDNA-sequencing showed a novel heteroplasmic transversion at nucleotide position 7671 in COII, responsible of the substitution of methionine to lysine [47].

In the same year, Clark KM et al. described a family with a heteroplasmic 7587TC mutation predicting the change of methionine to threonine. Particularly, the proband, who was the mother of the family, presented with unsteadiness of gait and fatigue, while her son struggled from a more severe condition starting at the age of five, with optic atrophy, pigmentary retinopathy, ataxia, and mild distal muscle atrophy [76].

Some years later, Horvath R et al., reported the case of a patient suffering from a reversible myopathy, lactic acidosis, exercise intolerance, and delayed growth with a heteroplasmic G9379A nonsense mutation (W58X) in the mtDNA encoding COIII subunit gene, while a G6708A nonsense mutation in the mtDNA COI gene encoding COX subunit I was identified by Kollberg G et al., in a 30-year-old woman with muscle weakness, pain, fatigue, and one episode of rhabdomyolysis [46,49].

Another form of reversible myopathy associated with COX deficiency was firstly described by Di Mauro et al. in 1983, but only after several years were characterized as attributable to a homoplasmic m.14674T>C or m14674T>G in the MT-TE gene coding the tRNA of the glutamic acid (see Section 4) [77,78].

Nevertheless, new pathogenic variants involving either mtDNA or nDNA genes coding for structural subunits, have emerged in recent years. A novel frame-shift variant in MTCO2 was recently observed in a 16-year-old girl with an infantile-onset history of exercise intolerance, in which whole-exome sequencing revealed a single base-pair deletion (m.8088delT) resulting in a premature stop codon [48].

A biallelic missense mutation was identified through whole exome sequencing in COX6A2, which is a COX-equipping subunit isoform expressed only in the skeletal muscle and heart, in two patients presenting with a congenital myopathy. The variants detected were homozygous c.117C > A (p.Ser39Arg) and compound heterozygous c.117C > A (p.Ser39Arg) and c.127T > C (p.Cys43Arg) [50].

Other symptoms evocative of skeletal muscle participation in the clinical picture are documented in some ancillary factor genes mutations. Mutations in these genes are usually associated with infantile onset with multisystem involvement and a fatal outcome.

SCO1 and SCO2 are metallochaperones that are essential for the assembly of the catalytic core of COX. Mutations in SCO2 cause fatal infantile encephalomyopathy and hypertrophic cardiomyopathy, whereas SCO1 patients presented with fatal infantile encephalomyopathy and hepatopathy [79]. Moreover, some cases of SCO2 mutations miming Werdnig–Hoffman syndrome are reported [80].

Finally, mutations in COX10 are rarely reported in adults with isolated COX deficiency, associated with a relatively mild clinical phenotype comprising myopathy; demyelinating neuropathy; premature ovarian failure; short stature; hearing loss; pigmentary maculopathy; and renal tubular dysfunction [51].

### 3.5. Complex V and Assembly Factors

ATP synthase, complex V, is the multimeric molecular enzyme responsible for the phosphorylation of ADP to obtain ATP. It consists of 13 different subunits and involves at least three ancillary factors. Several pathogenic mutations have been observed, especially in nDNA genes encoding complex V subunits (ATP5A1, ATP5E, USMG5) and in the ancillary factors (TMEM70 and ATPAF2) [81,82].

One of the most known genes is MT-ATP6, which encodes the complex V subunit “a” that contains the proton pore that releases the proton gradient established across the inner mitochondrial membrane. Since the discovery of MT-ATP6 mutation m.8993T>G, which was one of the first discovered [83], new pathogenic variants have been reported that result in a destabilization of the complex or in an impaired stability or dysfunction of the proton pumps, or in an increase ROS generation [84]. The most well-known syndrome is “NARP” (neuropathy, ataxia, and retinitis pigmentosa) syndrome due to the point mutation T8993G) in the gene encoding ATPase 6; the full clinical syndrome manifests when the percentage of mutated mtDNA compared to the total is 70–90% [85]. When the percentage of heteroplasmy exceeds 90%, the clinical manifestations resume the maternally inherited Leigh syndrome (MILS) [86]. The two syndromes frequently coexist in the same family, based on the level of mutation specifically present in each affected member [87].

In any case, mutations in the mitochondrial and nuclear genes coding for proteins and ancillary factors of complex V are mostly related to neuropathic involvement [88]. Muscle involvement is therefore unlikely to be the main clinical phenotype of these alterations, being mostly in association with other manifestations.

Sometimes muscle involvement may be misunderstood with other neuromuscular conditions.

Aurè et al. evidenced a homoplasmic deleterious mutation in the MT-ATP6/8 genes as potentially responsible for acute episodes of limb weakness mimicking periodic paralysis, which resolved positively with acetazolamide [89].

A case of MLASA (Mitochondrial Myopathy, Lactic Acidosis, and Sideroblastic Anemia) due to a de novo mutation (m.8969G>A) in the mitochondrially encoded ATP6 gene was described [90].

Moreover, Ganetsky et al. in a 2019 cohort study reported two individuals with exercise intolerance and a clinical picture resembling CPEO with two ATP6 variants (m.8608C>T predicting p.Pro28Ser and m.8723G>T predicting p. Arg66Leu) of uncertain significance [84].

## 4. Defect of Translational Apparatus

The translational apparatus consists of several components, including mt-tRNA, mt-rRNA, regulatory transcription factors, mitochondrial RNA polymerase, and mitoribosomes [91]. Particularly, the 22 mt-tRNAs play a fundamental role in the transport of amino acids to the developing polypeptide chain during the translation of mitochondrial proteins. If this process is altered, it can lead to a dysfunction of the ETC complexes I, III, IV. Over 200 variants have been described in patients with mitochondrial disorders with different phenotypes, but only a part of these variants fulfill the criteria to be defined “pathogenic” [92]. The resulting clinical phenotypes are extremely heterogeneous, including epilepsy, deafness, diabetes, ophthalmoparesis, myopathy, cardiomyopathy, and encephalopathy. Some classic syndromes such as MELAS and MERRF represent the most frequently observed. The well-known association of the tRNA Leu gene (MT-TL1) and MELAS as secondary to the m.3243A>G mutation dates back to the study by Kobayashi et al. in 1990 [93]. The m.3243A> G in the MT-TL1 gene is very specific, and it is still responsible for 80% of MELAS cases, followed by the m.3271T> C (representing 10% of cases) and the m.3252A> G (which is observed in less than 5%) [94]. MELAS syndrome affects several organs, and some of its manifestations include stroke-like episodes, dementia, epilepsy, lactic acidemia, myopathy, recurrent headaches, hearing impairment, diabetes, and short stature. Many patients carrying the 3243 variant do not manifest the entire symptomatology. Recently, an isolated myopathy was documented by Mahale RR et al., 2021, due to an m.3243A>G in the MT-TL1 gene [95]. Similarly, MERRF, which is usually secondary to the 8344A> G mutation in MT-TK, has been associated with other possible pathogenic alterations in different mt-tRNAs [96]. Beyond these typical clinical presentations, many pathogenetic variants involving mt-tRNAs have been described over the years as capable of leading to phenotypes with mainly myopathic presentations [97].

A myopathic clinical presentation has been reported in two patients with muscle weakness who were found to be carriers of the m.5631G>A and m.5610G>A mutations in the MT-TA gene [98].

Mt-tRNA defects manifested also with a CPEO phenotype, and several point mutations have been reported. A sporadic case of a heteroplasmic substitution in position 12316G>A in MT-TL2 causing cPEO with COX-negative fibers and RRF was described [99]; Karadimas CL et al. described an m.12315G>A mutation in the MT-TL2 gene in a woman with cPEO [100]; Soldath P et al. reported an m.12294 G>A in the MT-TL2 gene in an individual with cPEO and exercise intolerance [101].

More recently, an m.4414T>C in the MT-TM gene has been described as a variant causing PEO and myopathy in an adult patient whose muscle biopsy revealed focal cytochrome c oxidase deficiency and ragged red fibers [102]. Finally, a case of CPEO with a COX deficiency was also documented as due to mutation involving the MT-TN gene [103].

In other circumstances, mt-tRNAs defects may lead to a benign form. Fatigue, weakness, and exercise intolerance were the only clinical symptoms described in a family, in which all the 13 members presented isolated complex I deficiency and an m.3250T>C mutation in MT-TL1 [104].

A very interesting form of myopathy related to mt-tRNA mutation is a neonatal mitochondrial myopathy with COX deficiency, first described by Di Mauro et al. in 1983 [77]; a genetic definition was later reached in 2009 when Horvath et al. reported a case series of 17 individuals from 12 different families with a homoplasmic m.14674T>C or m14674T>G in the MT-TE gene coding the tRNA of the glutamic acid (Glu) [78,105]. This syndrome, known as benign reversible mitochondrial myopathy, affects mostly infants who struggle from lactic acidosis, limb myopathy and respiratory musculature involvement leading to respiratory failure. Fortunately, most of these patients spontaneously improve with supportive care [12]. The reason why these patients experienced a timed spontaneous recovery was not clear, but it is hypothesized that there is a developmental switch in the control strength of mitochondrial transfer RNAs and, in particular, MT-TE in mitochondrial translation, suggesting that 16–30% of steady-state levels of MT-TE may have a profound effect on translation in muscles of neonates, but this may become less critical at later stages of development.

Finally, it is worth mentioning that a myasthenia gravis-like phenotype due to a 5728T>C mutation in the MT-TN gene has been reported [106].

## 5. Defects of Electron Carriers

Coenzyme Q_10_ (CoQ_10_) is a lipophilic molecule comprising a quinone group and a 10-unit polyisoprenoid tail, that is located within the inner membrane of the human mitochondria. It functions as an electron carrier in the respiratory chain, transferring electrons from NADH:coenzyme Q reductase (complex I) and succinate: coenzyme Q reductase (complex II) to coenzyme Q:cytochrome c reductase (complex III). CoQ_10_ is also joined in pyrimidines, fatty acids and mitochondrial uncoupling proteins metabolism and serves as antioxidant [107].

CoQ_10_ deficiency has been reported in literature as a primary or secondary deficiency. The primary deficiency is associated with mutations of the at least nine genes involved in its biosynthesis, inherited in an autosomal recessive manner (COQ2, COQ4, COQ6, COQ7, CPOQ8A, COQ8B, COQ9, PDSS1, PDSS2) [108]. It can present with different phenotypes, usually characterized by multisystem involvement. Firstly, it was reported in 1988 by Ogasahara et al. who described two sisters with progressive muscle weakness, severe fatigability, and central nervous system dysfunction since early childhood [109]. Other neurologic manifestations include hypotonia, seizures, cerebellar features, myopathy, retinopathy or optic atrophy and sensorineural hearing loss [110].

A pure myopathic myopathy has been described with lipid storage myopathy and respiratory chain dysfunction [111,112]. Later, Gempel et al. found mutations in the ETFDH gene encoding electron-transferring flavoprotein dehydrogenase; seven patients from five families manifested exercise intolerance, fatigue, proximal myopathy, and hyperckemia. Muscle histology showed lipid storage and subtle signs of mitochondrial myopathy. All of the patients clinically improved after CoQ10 supplementation [113].

CoQ_10_ secondary deficiency is due to mutations in genes not directly involved in its pathway or to other environmental conditions.

Both CoQ_10_ primary and secondary deficiency can be improved by CoQ_10_ oral administration.

## 6. Defects of mtDNA Replication and Homeostasis

### 6.1. Rearrangements of mtDNA

mtDNA molecules’ replication and maintenance are strictly regulated by a complex apparatus. Mitochondrial diseases due to a defect in this machinery may manifest with heterogeneous phenotypes and different modalities of inheritance. Three types of rearrangements can be considered: singlelarge-scale mtDNA deletions, multiple large mtDNA deletions or mtDNA depletion; the latter two are the effects of defective mtDNA maintenance.

Large, single mtDNA deletion is mostly noninherited and characterized by a large nucleotide deletion from 1.3 to 7.6 kb [114]. The most “common” deletion is about 5 kb and spans from the ATPase 8 gene to the ND5 gene. The location varied within the mtDNA and is higher in heteroplasmy in the younger population. The mechanisms of producing single, large deletions during development remain unclear. Single, large deletions in mtDNA can be associated with three classic clinical syndromes: CPEO (chronic progressive external ophtalmoplegia), KSS (Kearns–Sayre syndrome) and Pearson syndrome [115]. A continuum of clinical phenotypes associated with single mtDNA deletions has been recognized. Age at onset and progression of disease seems to correlate with deletion size, heteroplasmy levels in skeletal muscle and location of the deletion within the mtDNA [116].

CPEO is the most benign form, and generally develops in third to the fourth decade and manifests with eyelid ptosis, ophtalmoplegia and myopathy. KSS manifest a multiystem involvement with more severe muscular impairment (weakness and wasting) retinopathy, ataxia, cardiac conduction defects, hearing loss, failure to thrive, ataxia and frequently abnormal brain MRI. The mean age at onset is about 20 years and prognosis is the worst. Cardiac conduction defects are frequent, and about 20% of these patients die of sudden cardiac death. PEO plus is a term frequently utilized in the clinical setting to identify patients with PEO and some degree of multisystem involvement.

Pearson syndrome is quite rare and severe; onset is during infancy, presenting as sideroblastic anemia and exocrine pancreatic dysfunction. Interestingly, the deletion is found in most tissues, suggesting that this event occurs very early in embryogenesis. Frequent clinical findings include short stature, cognitive impairment, sensorineural hearing loss, renal tubular acidosis, seizures, progressive myopathy, and endocrinopathies. Brain MRI showed cerebral and cerebellar atrophy and white matter changes. The disease is fatal in infancy [117].

### 6.2. Defects of mtDNA Mantainance

Mitochondrial DNA maintenance defects are a group of diseases caused by pathogenic variants in the nuclear genes involved in mtDNA maintenance, resulting in impaired mtDNA synthesis leading to quantitative (mtDNA depletion) and qualitative (multiple mtDNA deletions) defects in mtDNA.

The genes involved encode proteins belonging to at least three pathways: mtDNA replication and maintenance, nucleotide supply and balance, and mitochondrial dynamics and quality control (Figure 1) [118].

#### 6.2.1. Defects of mtDNA Replication Apparatus

The replication of mtDNA is continuous throughout the cell cycle in all cells. The apparatus needed for replication is exclusively encoded by nuclear genes.

mtDNA replication is managed by a specialized mtDNA polymerase, POL gamma. The polymerase is a heterotrimer composed of a catalytic subunit encoded by POLG (also reported in the literature as POLG1 and POLGA) and a homodimeric processing subunit composed of two p55 accessory proteins encoded by POLG2. Apart from POLG, the replisome consists of a helicase Twinkle (encoded by TWNK formerly named C10orf2), mitochondrial topoisomerase I, mitochondrial RNA polymerase, RNase H1 (encoded by RNASEH1), and mitochondrial genome maintenance exonuclease 1 (MGME1 encoded MGME1). Other proteins involved in mtDNA replication are mitochondrial single-stranded DNA binding protein 1 (mtSSBP1), DNA ligase III, DNA helicase/nuclease 2 (DNA2 encoded by DNA2), and RNA and DNA flap endonuclease (FEN1).

Defects in genes involved in the replication machinery may lead to mtDNA depletion, accumulation of multiple mtDNA deletions, or both, in critical tissues.

ANT1 encoded by SLC25A4, is the muscle-specific isoform of the mitochondrial adenine nucleotide translocator, and it is also expressed in the heart and brain. Several mutations in SLC25A4 have been linked to mitochondrial disorders and fall into two distinct clinical phenotypes: (1) Autosomal dominant CPEO [119,120] or (2) a slow progressive mitochondrial myopathy with cardiomyopathy characterized by fatigue and exercise intolerance and an autosomal recessive trait of inheritance [121].

Mutations of POLG are the most common pathogenic mutations and may present with a wide range of clinical presentations (https://tools.niehs.nih.gov/polg/index.cfm/polg, accessed on 30 November 2021). Three main syndromes can be recognized as (1) arPEO, usually characterized by isolated PEO and ptosis; (2) adPEO is frequently associated with myopathy and other systemic features; (3) ataxia neuropathy spectrum (ANS) combines the previous syndromes of mitochondrial recessive ataxia syndrome (MIRAS) and sensory ataxia neuropathy, dysarthria and ophthalmoplegia (SANDO), myoclonic epilepsy myopathy sensory ataxia (MEMSA) now envelops the syndrome of spinocerebellar ataxia with epilepsy (SCAE) and includes epilepsy, myopathy, and ataxia without PEO. Two other syndromic disorders related to POLG are recognized to present in childhood and include Alpers–Huttenlocher syndrome (AHS) and Childhood myocerebrohepatopathy spectrum (MCHS) disorder [122,123]

Autosomal dominant mutations in POLG2 have been shown to cause late-onset PEO with mtDNA deletions, but again, more complex phenotypes with ataxia, parkinsonism and neuropathy have been associated with POLG2 variants [124,125].

Heterozygous mutations in the *TWNK* gene are responsible for adCPEO [126] with accumulation of multiple mtDNA deletions. Clinical presentations include CPEO, often associated with proximal muscle and facial weakness, dysphagia and dysphonia, mild ataxia, and peripheral neuropathy. CPEO with parkinsonism has been rarely described in patients with twinkle mutations [127].

Mutations in MGME1, DNA2 and RNASEH1 have been reported in patients with PEO and accumulation of multiple mtDNA deletions. MGME1 and RNASEH1 mutations are inherited as recessive traits, whereas DNA2 defects seem to be dominant. Clinical manifestations appear in adulthood and more rarely in childhood, and in addition to PEO, there is involvement of respiratory muscles and the brain, with cerebellar atrophy [128,129,130].

#### 6.2.2. Defects of Mitochondrial Deoxyribonucleosides Pools

Mitochondrial deoxynucleotides triphosphate (dNTP) pools depend on two different pathways, the de novo synthesis due to an active transport of cytsolic dNTPs from reduction of ribonucleotides by ribonucleotide reductase (RNR) and the salvage pathway through the purines and pyrimidines by action of two mitochondrial deoxyribonucleoside kinases, thymidine kinase 2 (TK2) and deoxyguanosine kinase (DGUOK). In nondividing cells, cytosolic TK1 and dNTP synthesis is downregulated, forcing the burden of mitochondrial dNTP pool synthesis on the two mitochondrial deoxyribonucleoside kinases.

Mutations in the genes for enzymes involved in both pathways cause several forms of mtDNA depletion syndromes (Table 4) [131].

MDS are autosomal recessive disorders with a broad genetic and clinical spectrum.

The salvage pathway is essential in postmitotic cells such as neurons and muscle cells, in which dNTPs are produced by utilizing preexisting nucleosides through a complex pathway of enzymes [132]. Among them, there are enzymes encoded by SUCLA2 (adenosine diphosphate (ADP)-forming succinyl CoA ligase beta subunit), SUCLG1 (guanosine diphosphate (GDP)-forming succinyl CoA ligase alpha subunit) and TYMP (thymidine phosphorylase) [133].

**Table 4 jcm-11-00632-t004:** Examples of defects in nuclear genes with a predominant muscular involvement phenotype.

Genes	Type of Article	Patients/Age at Onset	Muscle Manifestations	Other Clinical Features	References
SUCLG1	CR	1 pt 6mo	Marked muscular hypotonia, severe muscle atrophy	Failure to thrive	[134]
SUCLG1	CR	1 pt 17 h	Axial hypotonia, no head control, poor swallowing and muscle weakness	Severe metabolic acidosis, liver failure	[132]
SUCLA2	CS	50 pts Median age: 2 mo	Eyelid ptosis, ophthalmoplegia	Psychomotor retardation and failure to thrive, dystonia, hearing impairment, epilepsy	[132]
TK2	CS	92 pts Age range: birth–72 yo	Proximal muscle weakness, high CK, ptosis, PEO, dysphagia, dysarthria/dysphonia,	seizures, encephalopathy and cognitive impairment, sensorineural hearing loss	[135]
DGUOK	CS	6 pts Age range: 20 yo–69 yo	PEO, ptosis, limb girdle weakness, myalgia, dysphonia, dysphagia		[129]
DGUOK	CR	1 pt 14 yo	Fatigue during exercise	Abdominal pain, reduced left ventricular systolic	[136]
TYMP	CS	102 pts Age range: 11–59 yo	Bilateral ptosis, ophthalmoparesis, lower limb hyposthenia	Vomiting, abdominal pain, severe malnutrition	[14]
RRM2B	CS	3 pts Age range: 8 we–27 mo	Muscle hypotonia and progressive weakness, poor head control and respiratory distress. High CK	Failure to thrive and diarrhea, persistent acidosis, sensorineural hearing loss, retinopathy	[137]
RRM2B	CS	31 pts Age range: 0–6 mo	Muscular hypotonia, respiratorydistress	Failure to thrive, hearing loss, encephalopathy, seizures. Renal, eye and GI involvement. Anemia	[138]
POLG	CS	95 pts Age range: 2–23 yo	Myopathy, ptosis and PEO, neck flexor weakness	Seizures, hepatopathy, lactic acidaemia, sensory ataxia, bradykinesia	[123]
Twinkle	CS	4 pts Age range: 20s–30s	Ptosis and ophthalmoplegia, diplopia	Gait difficulty, stiffness, resting tremor, depression	[127]
SLC25A4	CS	25 pts Age range: 0–48 yo	Myopathy and muscle hypotonia, dysarthria, respiratory insufficiency	Ischemic stroke, hydrocephalus, insomnia, mental retardation, headache, cardiomyopathy, cataracts, scoliosis	[121]
MGME1	CS	6 pts Age range: NA	PEO, proximal weakness, generalized muscle wasting, respiratory failure, exercise intolerance	Mental retardation, depressive episodes, gastrointestinal symptoms, spinal deformities, ataxia, dilated cardiomyopathy and arrhythmias,	[128]
OPA 1	CS	8 pts Age range: birth–60 yo	Ptosis and ophthalmoplegia, exercise intolerance and myalgia, muscle weakness	Optic atrophy, hearing loss, pes cavus, feeding difficulties	[139]
SPG7	CS	9 pts Age range: Late20s–Mid60s	Ptosis and ophthalmoplegia, proximal myopathy, dysphagia, dysphonia, dysarthria	Spasticity, ataxia, bladder symptoms, mild cognitive impairment	[140]

F: female; M: male; NA: not available yo: years old; CR: case report; CS: cohort study.

SUCLA2 and SUCLG1 genes encode subunits of succinyl CoA ligase (SUCL), a mitochondrial enzyme of the Krebs cycle that catalyzes the reversible conversion of succinyl-CoA and ADP or GDP to succinate and ATP or GTP [134].

SUCL deficiency is an MDS, presenting as Leigh/Leigh-like encephalomyopathy with variable phenotypes. It occurs in early childhood with severe hypotonia and failure to thrive, then developing into muscular atrophy and growth retardation. Other clinical features could be dystonia, sensorineural hearing impairment and respiratory problems. Basal ganglia lesions, cerebral atrophy and white matter alterations are common findings at neuroimaging as well as lipid accumulation and type I fibers’ predominance at muscle biopsy. Additionally, it is variably associated with methylmalonic aciduria, because accumulated succinyl-CoA is subsequently converted to methylmalonyl-CoA [132].

TK2 catalyzes the conversion of deoxycytidine and thymidine nucleosides to their nucleoside monophosphates (dCMP and dTMP), that are ready to be phosphorylated to dNTP and incorporated into mtDNA. TK2 deficiency presents predominantly as a proximal myopathy with variable severity, leading in most cases to loss of ambulation and respiratory insufficiency in a few years. Three main clinical forms have been described: a rapidly progressive infantile-onset myopathy, associated with CNS alterations such as encephalopathy, seizures or cognitive impairment; a childhood-onset myopathy, resembling SMA type 3; and a late-onset myopathy with usual facial and extraocular muscles involvement (i.e., ptosis or PEO) [135].

To date, therapy for TK2 deficiency is still at a preclinical status. It consists of replacement of dCMP and dTMP or their respective nucleosides deoxycytidine (dC) and deoxythymidine (dT), directly. They were administered orally in TK2−/− mice, resulting in reduced imbalance of mtDNA, recovery of mitochondrial functions and prolonged lifespan [141].

DGUOK gene encodes for deoxyguanosine kinase (dGK), an enzyme that plays the same role as TK2 in the purine nucleoside salvage pathway. It converts deoxyguanosine and deoxyadenosine to, respectively, deoxyguanosine (dGMP) and deoxyadenosine monophosphate (dAMP), that need another two phosphorylation steps before being inserted into mtDNA.

dGK deficiency is mostly known as a neonatal onset multisystem disease, characterized by hepatic and neurologic dysfunction such as hypotonia, psychomotor retardation, and typical rotary nystagmus. Few individuals are affected later in infancy or during childhood, when the disease presents as isolated hepatic disease [142].

Ronchi et al., 2012, reported five patients with dGK deficiency and variable skeletal muscles’ involvement (progressive external ophthalmoplegia and ptosis, dysphonia and dysphagia, limb girdle weakness, myalgia, cramps and rhabdomyolysis) [129].

Finally, Buchaklian et al., 2012, described a case of juvenile-onset myopathy presenting with weakness and fatigability [136].

Notably, recent studies have demonstrated that the nucleoside salvage pathway is coadjuvated by the nuclear triphosphohydrolase enzyme SAMHD1. Its role is to hydrolyze in the nucleus dNTPs synthesized de novo in the cytosol. The obtained deoxynucleosides can be recycled by TK2 and dGK for mtDNA replication [143].

TYMP gene encodes for thymidine phosphorylase (TP), which catabolizes thymidine (dThd) and deoxyuridine (dUrd) into their respective bases. Lack or dysfunction of TP determines dThd and dUrd toxic accumulation with consequent mtDNA impairment, leading to Mitochondrial Neuro–Gastro–Intestinal Encephalomyopathy (MNGIE) [144].

MNGIE is an ultrarare condition, which can present as “Early Onset” (or “Classic”) and “Late Onset” forms. Main clinical features are represented by gastrointestinal symptoms (diarrhea, abdominal pain, pseudo-obstruction, weight loss/cachexia) and neurological symptoms/signs (ptosis, ophthalmoparesis, polyneuropathy, hearing loss and leucoencephalopathy at brain MRI imaging), with fatal evolution. Several therapeutic options have been proposed to replace TP temporarily, for example, erythrocyte-encapsulated TP infusions, or to restore TP permanently such as hematopoietic stem cell transplantation and orthotopic liver transplantation. Recently, gene therapy was successful in restoring biochemical homeostasis in a murine model of the disease [14,145,146].

Another gene involved in mitochondrial nucleoside salvage is 4-aminobutyrate aminotransferase (ABAT), that encodes for GABA transaminase (GABA-T). This enzyme is responsible for both GABA neurotransmitter catabolism in the mitochondrial matrix and conversion of dNDPs to dNTPs. Thus, ABAT deficiency causes GABA accumulation and mitochondrial nucleoside pools imbalance [147]. The resulting MDS usually occurs in neonatal age with hypotonia and severe psychomotor retardation. Visual impairment, abnormal movement, hypersomnia and infantile-onset refractory epilepsy have been reported, too [148].

In contrast to the mitochondrial salvage pathway, nucleotide precursors can be directly obtained through reduction in ribonucleoside diphosphates to deoxyribonucloside diphosphates by ribonucleotide reductase. This enzyme is composed of R1 and R2 subunits; the last one is known as p53-inducible form (p53R2) and plays a crucial role in de novo nucleotide synthesis [149].

It is encoded by RRM2B gene, whose mutations can cause an encephalomyopathic form of MDS, which is clinically heterogeneous. Commonly, RRM2B deficiency presents as neonatal or infantile myopathy with lactic acidosis and increased serum CK. Therefore, children with combinations of hypotonia, tubulopathy, seizures, respiratory distress, diarrhea and lactic acidosis have been described [137].

Other less common RRM2B phenotypes have been described as progressive external ophthalmoplegia with bulbar dysfunction, fatigue, and muscle weakness with autosomal dominant transmission in adults or with a recessive trait and childhood-onset [138,150,151].

#### 6.2.3. Defects of Mitochondrial Dynamics and Quality Control

Mitochondria are highly dynamic organelles, providing themselves with different shapes, distributions and sizes though fusion and fission reactions. The related enzymatic machinery is called “mitochondrial dynamics” and defects in its components can be associated with various disorders [15].

Mitochondrial fusion is the merger of two mitochondria into one, based on a GTPases-mediated process. Optic atrophy 1 (OPA1) is involved in the outer membrane fusion, while GTPases for the inner membrane fusion are Mitofusins 1 and 2 (MFN1 and MFN2) [152]. Mitochondrial fission is the opposite process—the division of a mitochondrion into two smaller mitochondria. The main protein of the fission is dynamin-related protein 1 (Drp1), a GTP-hydrolyzing enzyme [15,152].

The large majority of mutations in the OPA1 gene are related with a slowly progressive optic neuropathy. Dominant optic atrophy (DOA) is due to OPA1 mutations in about 60–70% of cases [139,153]. It is caused by the degeneration of optic nerve fibers with mild visual loss and color vision alterations, usually starting during childhood. In about 20% of patients, DOA evolves as syndromic forms with myopathy, progressive external ophthalmoplegia, peripheral neuropathy, stroke, multiple sclerosis, spastic paraplegia and sensorineural hearing loss [154].

MNF1 forms homomultimers and heteromultimers with MFN2. No human diseases have been described in relation to MFN1 mutations [16]. Mutations in MFN2 are associated with autosomal dominant or recessive Charcot–Marie–Tooth disease type 2A and autosomal dominant hereditary motor and sensory neuropathy VIA. Charcot–Marie–Tooth disease type 2A is characterized by distal limb muscle weakness and atrophy, axonal degeneration/regeneration, areflexia and distal sensory loss, associated variably with CNS involvement, optic atrophy, hearing loss and vocal cord paresis [15].

Another protein located at the inner mitochondrial membrane is paraplegin, encoded by the SPG7 gene as part of the AAA family of ATPases. It contributes to assembly of mitochondrial ribosomes, taking part in other proteins’ synthesis [155]. SPG7 mutations have been recently linked to PEO mid-adult onset, with multiple mitochondrial DNA deletions. Ptosis, spastic ataxia, dysphagia and proximal myopathy were other common features [140].

## 7. Genetic Diagnostic Approach in PMM

In clinical practice, despite the increased awareness of mitochondrial disorders and the introduction of deeper genetic investigations, the diagnosis, in a patient with a suspected mitochondrial disease remains a challenge for clinicians (Figure 2). The main muscle symptoms are characterized by a frequent involvement of extraocular muscles with eyelid ptosis and PEO associated with exercise intolerance and muscle weakness of varying degrees at four limbs. Most often muscle involvement is in the context of a multisystem clinical presentation. 

The first step in the diagnostic process is an accurate evaluation of family history. 

The presence of a sporadic case or a matrilinear inheritance evoke an mtDNA-related disorder, and mtDNA testing is suggested. In sporadic cases, according to the data of frequency coming from the International Registries [115,156], a large mtDNA rearrangement should be searched for, but it is not meaningless to consider that testing for pathogenic mtDNA variants in blood alone can yield false-negative diagnoses, and the use of other samples (e.g., urinary sediment, buccal swab) can improve the sensitivity of diagnostic mtDNA testing.

Targeted NGS panels can be applied to reduce time for genetic diagnosis, but it is clear that WES or WGS approaches are more powerful. Both generate vastly more data and require additional analysis. These approaches can simultaneously analyze both genomes, can identify unknown variants and novel disease genes.

Interpretation of genomic sequencing results has been greatly improved by adoption of the American College of Medical Genetics criteria for variant classification, but requires a multidisciplinary team comprising clinicians with expertise in mitochondrial disorders, to evaluate the consistency with the phenotype.

In addition, unknown or uncertain variants need to be tested by functional testing using more traditional approaches for diagnosis including morphological and biochemical studies.

## 8. Conclusions

Although the heterogeneity and pleiotropy of mitochondrial disorders are not sufficiently clarified, over recent decades the introduction of broad-based exome sequencing as the standard first-line diagnostic approach has increased the yield of definite diagnosis in patient with a suspected mitochondrial disorders. Nowadays, the identification of the genetic basis of disease in each patient is relevant, particularly among patients with a PMM that is becoming the main target phenotype in clinical trials.

## Figures and Tables

**Figure 1 jcm-11-00632-f001:**
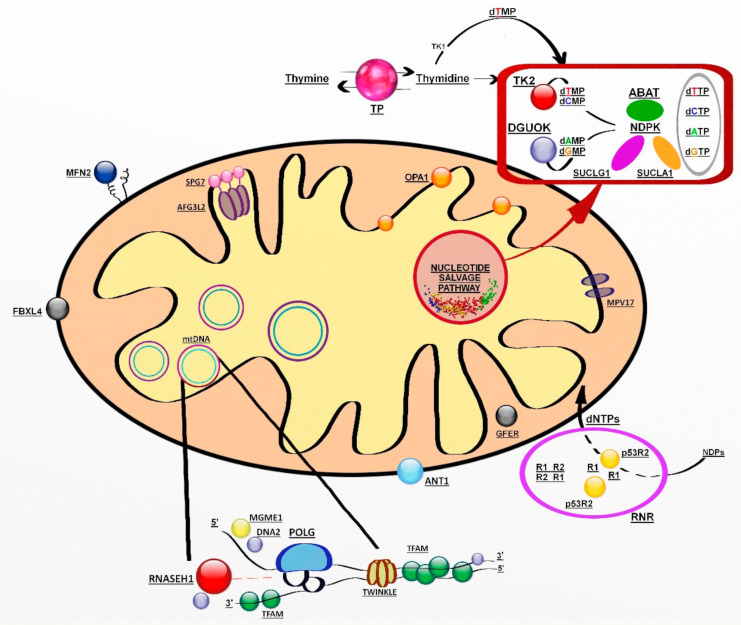
Genes involved in mitochondrial DNA maintenance. Abbreviations: ABAT, 4-aminobutyrate aminotransferase; AFG3L2, AFG3-like protein 2; ANT1, adenine nucleotide translocator 1; DGUOK, deoxyguanosine kinase; DNA2, helicase/nuclease DNA2; dATP-dCTP-dGTP- dTTP (deoxynucleoside triphosphates); dAMP-dCMP-dGMP-dTMP (deoxynucleoside monophosphates); FBXL4, F-box/LRR-repeat protein 4; GFER, growth factor, augmenter of liver regeneration; MFN2, mitofusin 2; MGME1, mitochondrial genome maintenance exonuclease 1; MPV17, protein MPV17; NDPK, nucleoside diphosphate kinase, OPA1, dynamin-like 120 kDa protein; POLG catalytic subunit of polymerase gamma; p53R2; p53-inducible small subunit of the ribonucleotide reductase; R2, small subunit of the ribonucleotide reductase; RNASEH1, ribonuclease H1; RNR, ribonucleotide reductase; SPG7, paraplegin; SUCLA2, β-subunit of the succinate-CoA ligase; SUCLG1, α-subunit of the succinate-CoA ligase; TFAM, mitochondrial transcription factor 1; TK2, thymidine kinase 2; TP, thymidine phosphorylase; TS, thymidylate synthase; Twinkle, mitochondrial helicase.

**Figure 2 jcm-11-00632-f002:**
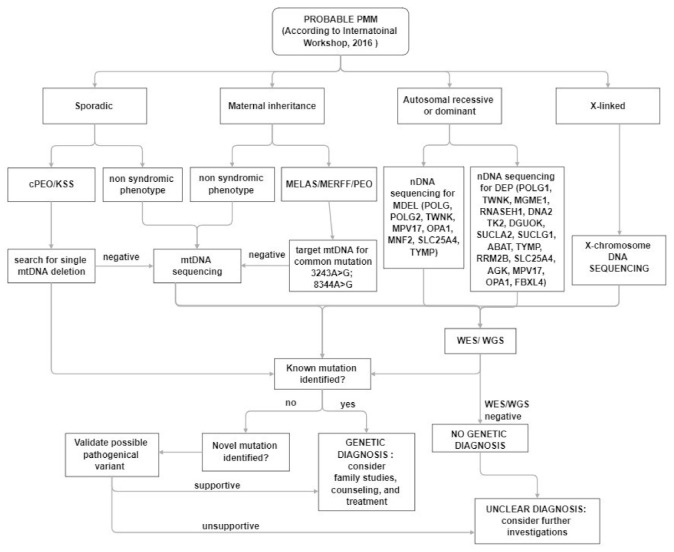
Diagnostic algorithm for patients with suspected PMM.

**Table 1 jcm-11-00632-t001:** Type of mitochondrial dysfunction and main PMM phenotypes.

Mitochondrial Dysfunction	Involved Gene	Muscle Phenotype
**Defects of MRC Complexes**		
Complex I	Mitochondrial-encoded subunits: Mt-ND1, Mt-ND2, Mt-ND3, Mt-ND4, Mt-ND4L, Mt-ND5, Mt-ND6 Nuclear-encoded subunits: NDUFA1, NDUFA2, NDUFA9, NDUFA10, NDUFA11, NDUFA12, NDUFA13, NDUFB3, NDUFB9, NDUFB10, NDUFB11, NDUFS1, NDUFS2, NDUFS3, NDUFS4, NDUFS6, NDUFS7, NDUFS8, NDUFV1, NDUFV2 Assembly factors: ACAD9, FOXRED1, NDUFAF1, NDUFAF2, NDUFAF3, NDUFAF4, NDUFAF5, NDUFAF6, NUBPL, TIMMDC1, TMEM126B	Isolated myopathy, multisystem involvement, MELAS
Complex III	Mitochondrial-encoded subunits: Mt-CYB Nuclear-encoded subunits: CYC1, UBCRB, UQCRC2 Assembly factors: BCSIL, LYRM7, TTC19, UQCC2	Isolated myopathy, multisystem involvement
Complex IV	Mitochondrial-encoded subunits: Mt-CO1, Mt-CO2, Mt-CO3 Nuclear-encoded subunits: COX41, COX412, NDUFA4 Assembly factors: COA3, COA5, COA6, COA7, COX10, COX14, SCO1, SCO2, COX15, COX20, PET100, APOFT1, SURF1, PET 11	Isolated myopathy, multisystem involvement
Synthesis of electron carriers		
	COQ2, COQ4, COQ5, COQ6, COQ7, COQ8A, COQ8B, COQ9, PDSS1, PDSS2 CYCS, HCCS	Isolated myopathy, nephropathy, cardiomiopathy
mtDNA replication and maintenance		
mtDNA homeostasis	DNA2, MGME1, POLG, POLG2, RNASEH1, TWNK	cPEO, cPEO plus
Maintenance of mitochondrial nucleotide pools	ABAT, DGUOK, MPV17, RRM2B, SAMHD1, SUCLA2, SUCLG1, TK2, TYMP	cPEO, cPEO plus, isolated myopathy, MNGIE, MDDS
Disorders of mitochondrial dynamics and quality control		
	Mitochondrial Membrane Phospholipid Metabolism andProtein Import Machinery TAZ, TIMM8A XL AGK, CHKB, DNAJC19, GFER, PAM16, SERAC1, PLA2G6, TIMM22, TIMM50, TIMMDC1 Mitochondrial Membrane Phospholipid Metabolism and Protein Import Machinery TAZ, TIMM8A XL AGK, CHKB, DNAJC19, GFER, PAM16, SERAC1, PLA2G6, TIMM22, TIMM50, TIMMDC1 DNM1L, MFN2, OPA1, GDAP1, MSTO1 AD/AR MFF, STAT2, TRAK1, MIEF2 DNM1L, MFN2, OPA1, GDAP1, MSTO1, AFG32, SPG7	cPEO

**Table 2 jcm-11-00632-t002:** Examples of reviews focusing on different aspects of mitochondrial disorders in the last decade.

Title	Main Focus	Reference
Mitochondrial disease in adults	Clinical aspects and diagnosis	[2]
Mitochondrial energy generation disorders: genes, mechanisms, and clues to pathology.	Genetic discovery and functional characterization	[3]
Mitochondrial Disease Genetics Update Recent insights into the Molecular Diagnosis and Expanding Phenotype of Primary Mitochondrial Disease.	Update of novel mitochondrial disease genes and pathogenic variants	[4]
Mitochondrial disease in adults: what’s old and what’s new?	Disease mechanism and clinical aspects in adults	[5]
Mutations causing mitochondrial disease: What is new and what challenges remain?	Advances in mitochondrial genetics	[6]
Human diseases associated with defects in assembly of OXPHOS complexes.	Factors involved in assembly human OXPHOS complex	[7]
Complex I deficiency: clinical features, biochemistry and molecular genetics.	Advances in the structure, function and assembly of complex I	[8]
The genetics and pathology of mitochondrial disease.	Genetic discovery and advances in mitochondrial pathology	[9]
Nuclear gene mutations as the cause of mitochondrial complex III deficiency	Discuss the nuclear-encoded proteins in which mutations have been found to be associated to CIII deficiency	[10]
Cytochrome c oxidase deficiency	Genetic etiology and clinical manifestations in COX deficiency	[11]
Mitochondrial disease in children.	Clinical aspects and diagnosis	[12]
Mitochondrial DNA depletion syndromes: review and updates of genetic basis, manifestations, and therapeutic options.	Genetic basis, clinical manifestation, and therapeutic options	[13]
Clinical and genetic spectrum of mitochondrial neurogastrointestinal encephalomyopathy	Symptomatology, diagnostic procedures, and hurdles, in vitro and in vivo models, experimental therapies	[14]
Mitochondrial dynamics: overview of molecular mechanisms	Overview of the molecular mechanisms that govern mitochondrial fission and fusion in mammals.	[15]
Nuclear genes involved in mitochondrial diseases caused by instability of mitochondrial DNA	Overview of nuclear genes involved in mitochondrial diseases	[16]

**Table 3 jcm-11-00632-t003:** Examples of genetic alterations of isolated MRC Complex I to V deficiency with a predominant muscular involvement phenotype (or isolated mitochondrial myopathy).

Genes	Type of Article	Type of Mutation	Patients N/Age at the Time of the Examination	Clinical Presentation	References
MT-ND1	CR	m.4087A>G (pThr261Ala) in homoplasmy	1 pt/41 yo	Ptosis, ophthalmoparesis, weakness, rabdomyolisis after suxamethonium injection	[31]
MT-ND1	CR	m.3365T>C (pLeu20Pro)	F/28 yo	Exercise intolerance, fatigue, weakness. Metabolic acidosis, high serum lactate concentration	[32]
MT-ND1	CR	m.4175G>A (pTrp290 *)	NA/22 yo	Exercise intolerance, exertion-related muscle weakness and pain. Ptosis. High serum lactate concentration	[32]
MT-ND1	CR	m.3902_3908invACCTTGC	M/43 yo	Exercise intolerance and myalgia. Weakness.	[36]
MT-ND2	CR	m.4831G>A (pGly121Asp)	M/21 yo	Fatigability, muscle weakness. High serum lactate concentration.	[33]
MT-ND2	CR	m.5133_5134del	M/28 yo	Exercise intolerance. High serum lactate concentration	[20]
MT-ND4	CR	m.11832G>A (pTrp358 *)	M/38 yo	Exercise intolerance, fatigue, myalgia.	[34]
MT-ND5	CR	m.13271T>C (p. Leu312Pro)	F/27 yo	Exercise intolerance. High serum lactate concentration.	[35]
MT-ND6	CR	m.14512_14513del p.(Met54Serfs *7)	M/27 yo	Exercise intolerance, muscle weakness, ptosis, intermittent diplopia.	[38]
TMEM126B	CR	c.635G>T (p.Gly212Val) and c.401delA (pAsn134Ilefs *2)	2F and 4M/ 21–36 yo	Exercise intolerance, muscle weakness. High serum lactate concentration	[39]
TMEM126B	CR	c.635G>T (p.Gly212Val) and c.397G>A (p.Asp133Asn) (2/3); c.635G>T (pGly212Val) and c.208C>T (p.Gln70 *)	1F and 2M/ 22–38 yo	Proximal muscle weakness. Exercise induced myalgia. High serum lactate concentration	[40]
ACAD9	CS	42 missense mutations,	41F and 29M/22 days—44 yo	Exercise intolerance, muscle weakness, cardiomyopathy. Metabolic acidosis. High serum lactate concentration	[41]
		1 frame shift, 1 nonsense,			
		7 splice sites and 1 initiation codon			
MT-CYB	CR	m.15084G>A (pTrp113 *);	2F and 3M/ 32—52 yo	Exercise intolerance, muscle weakness. High serum lactate concentration. Exercise-induced myoglobinuria.	[34]
		m.15168G>A (pTrp141 *);			
		m.15723G>A (pTrp326 *);			
		m.14846G>A (pGly34Ser);			
		m.15498_15521del			
MT-CYB	CR	m.15615G>A (pGly290Asp)	M/29 yo	Exercise intolerance	[42]
MT-CYB	CR	m.15242G>A (pGly166 *)	F/34 yo	Exercise intolerance. High serum lactate concentration; encephalopathy with seizures and hallucinations.	[43]
MT-CYB	CR	m.1517G>A (pGly142 *)	F/40 yo	Exercise intolerance, muscle weakness, fatigue and myalgia	[44]
MT-CYB	CR	m.15800C>T (pGln352 *)	F/24 yo	Exercise intolerance, fatigue.	[45]
MT-CO1	CR	m.6708G>A (predicting the loss of the last 245 amino acids of 514 in COX II)	F/30 yo	Exercise intolerance. Muscle weakness and fatigue, myalgia. Exercise-induced myoglobinuria.	[46]
MT-CO2	CR	m7671T>A p(Met29Lys)	M/14 yo	Muscle weakness and fatigue. High serum lactate concentration.	[47]
MT-CO2	CR	m8088delT (pLeu168 *)	F/16 yo	Exercise intolerance, muscle weakness. High serum lactate concentration. Anemia	[48]
MT-CO3	CR	m9379G>A (pTrp58 *)	M/20 yo	Exercise intolerance. Muscle weakness, hypotonia and scapular winging; The symptoms spontaneously regressed through the years.	[49]
COX6A2	CR	c.117C>A (p.Ser39Arg) and c.127T>C (p.Cys43Arg)	M/9 yo	Muscle weakness, hypotonia, facial weakness, high arched palate since infancy	[50]
COX10	CR	c.1007A>T (p.Asp336Val) and c.1015C>T (p.Arg339Trp)	F/37 yo	Failure to thrive. Exercise intolerance, muscle weakness and fatigue, renal Fanconi syndrome. Metabolic acidosis.	[51]

F: female; M: male; NA: not available yo: years old; * truncation; CR: case report; CS: cohort study. All the mtDNA mutations were heteroplasmic except for one (Rafiq J et al., 2015).

## Data Availability

Not applicable.

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
