# Peer review of "Molecular Genetics Overview of Primary Mitochondrial Myopathies"

_jcm, 2022, doi:10.3390/jcm11030632_

Round 1

Reviewer 1 Report

The authors set out to review the many mitochondrial mutations which cause a human phenotype. Despite the large number of rare mutations the authors have succeeded in their endeavor. I find this review well written and timely. The manuscript needs one more proof-reading, for example:

3.3 second paragraph - complex III represents the center...

3.3 sixth paragraph - sequencing was it possible....

3.3 ninth paragraph - identified such as ataxia...

Reviewer 2 Report

This current review, written by Evora et al. gives an overview of Molecular Genetics defects in Primary Mitochondrial Myopathies. Overall, the flow of the manuscript is good, with a general organization of the manuscript making sense, first introducing their literature research approach (and choices) accompanied by some general aspects of mitochondrial genetics. Then, in the following parts of the manuscript, the authors present the different mitochondrial molecular defects and their consequences on muscular phenotype illustrated by two tables and two figures.

Major Comments:

The main topic of the review (i.e Primary Mitochondrial Myopathies) has generated a significant number of publications including many review articles with at least 50 reviews dealing with primary Mitochondrial Myopathy during the last 10 years. I suggest the authors narrow more the topic by recapitulating the recent knowledge brought by these previous reviews in the introduction section. It could be interesting to summarize the relevant reviews in a table. This would be of great help for the reader to easily gets the authors evaluation and choices and clearly highlights new information provided by the present work.

The organization of table 1 should be revised and/or completed. Although tables should summarize what's mentioned in the main text, they should also stand-alone and be self-explanatory. When reading table 1, it’s not possible to see from which reference(s) informations come from. Hence, table 1 should be more informative and would benefit from at least the addition of an extra- column with literature references.

Section 7 dealing with the diagnostic approach is hard to read and understand. It notably contains a truncated sentence and undefined abbreviations like WES and WGS in the second paragraph. Hence, it should be rewritten.

Other comments:

The reference list, the figures and tables should be carefully checked. Some references are either missing or duplicated. For instance, Duplicated references were found 69 and 101 Dimauro et al, 79 and 86 Ganetzky et al., Horvath et al., ref 70 and 102, ref 2 and 161 Ng et al., ref 128 and 138 Ronchi et al.

When citing a review, the authors should clearly indicate the readers that the information is coming from a review article.

The manuscript should be proofread for some typos, grammar errors, redundant statements, inconsistencies of format (includingformat of reference citations, or missing commas or points). For instance :

page 7 : by Horvarth et al in the 2009.”

page 10 :  hypnotized instead of hypothesized.

page 12 : a line break is introduce in the middle of the sentence starting with  Defect in genes involved…

page 13 : missing full stop at the end of the first sentence

Reviewer 3 Report

It is a well-written review.

2, general aspect

Please change: “1 gene encode----” to “One gene encode---”

                        “3 genes encode---” to “Three gene encode----"

                                “majority is synthetized starting from nDNA” to “majority is encoded nDNA”

3.1. Complex I and assembly factors

“Complex I (CI) also known as NADH-coenzyme Q oxidoreductase or NADH dehydrogenase,” Please delete “NADH dehydrogenase” in that NADH dehydrogenase is only part of complex I.

3.2. Complex II and assembly factors

“Complex II (also called succinate dehydrogenase) is ----” Please change to “Complex II (also called succinate-coenzyme Q oxidoreductase” in that succinate dehydrogenase is only part of complex II.

3.3. Complex III and assembly factors

“CYC as well the other” please change to “CYC1 as well the other”

3.4. Complex IV and assembly factors

“since NDUFA4, previously assigned to complex I, was recently added as a new peripheral subunit of COX [62]” Could authors make it clear that NDUFA4 only belongs to COX4 or NDUFA4 belongs both complex I and complex 4 now?
